# The intentions of pregnant women to give birth at a health facility and associated factors in the Aleta-Wondo rural District, Ethiopia: A community based cross-sectional study

Aregahegn Dona◉*, Azmach Dache Mue

Department of Social and Population Health, Yirgalem Hospital Medical College, Yirgalem, Sidama, Ethiopia

* aregahegndona@gmail.com

**Data Availability Statement:** All relevant data are within the paper and its Supporting Information files.

## Abstract

Pregnancy and childbirth-related complications are the leading causes of death among women of the reproductive age group. Giving birth at a health facility is crucial to prevent these complications. Hence, this study aimed to assess the pregnant women's intentions to give birth at a health facility and associated factors in the Aleta-Wondo rural District, Ethiopia. A community-based cross-sectional study was conducted among randomly selected 421 pregnant women. Data were collected by using an interviewer-administered structured questionnaire. The collected data were entered into Epidata 3.1 and exported to SPSS version 21 for analysis. Bivariable and multivariable logistic regression analyses were done. An adjusted odds ratio with a 95% confidence interval was used to assess the presence and strength of association. A p-value ≤0.05 was applied to declare statistical significance. Generally, 61.3% (95% CI: 57.0, 66.3) of the respondents intended to give birth in a health facility. Receiving information from health professionals (AOR = 2.6; 95% CI: 1.5, 4.4), perceived threats (AOR = 4.5; 95% CI: 2.6, 7.6), perceived benefits (AOR = 2.3; 95% CI: 1.1, 4.9), perceived barriers (AOR = 0.4; 95% CI: 0.3, 0.7) were factors significantly associated with pregnant women's intention. Pregnant women's intention to give birth in a healthcare setting is low in the study area. Strengthening information communication with healthcare professionals and reducing threats and barriers that affect pregnant women's intentions is essential. Moreover, we recommend further research with mixed methods.

## Introduction

Pregnancy and childbirth-related complications are the leading causes of death and disability among women of the reproductive age group [1]. Giving birth at a health facility can save nearly three-fourths of maternal deaths [2].

Therefore, the choice of place of delivery for a pregnant woman is an important aspect of maternal healthcare [3]. However, many beliefs and misconceptions about pregnancy and childbirth influence women's intention to choose a place of birth, which in turn affects the health outcomes of both the mother and the baby [4].

**Funding:** The authors received no specific funding for this work.

**Competing interests:** The authors have declared that no competing interests exist.

Despite the agreement that access to healthcare must be universal and guaranteed for all on an equitable basis, women continue to face significant inequities in using available healthcare services, particularly in low-income countries [5]. In many low-income countries including Ethiopia, pregnancy and childbirth are often supposed as normal life events without justification to seek skilled help [6]. The intentions of pregnant women can be influenced by their attitudes, perceived ability to perform a behavior, and subjective beliefs regarding that behavior [7].

Even though maternal death has declined worldwide, this reduction has been highly variable, with low-income countries owning the largest burden [8]. From low-income countries, Sub-Saharan Africa alone contributes to almost two-thirds of these deaths [8]. Pregnant women's ignorance of receiving skilled care is one of the main contributing causes to these deaths [7, 8]. In spite of improvements in the availability and accessibility of healthcare services in Ethiopia, only 40% of live births were delivered in a health facility in rural areas [9].

The pregnant women's intentions can be influenced by their perceptions towards childbirth-related complications; they may prepare to take preventive actions when the threat perception towards these complications is high, [10, 11]. In addition, pregnant women who perceive the need for professional help and recognize the risk of childbirth-related complications will be more interested in delivering at a health facility [12].

In fact, it is challenging to predict and detect life-threatening conditions during pregnancy and childbirth unless managed by skilled healthcare providers [13, 14]. Giving birth at a healthcare facility is the best way to ensure a safe and successful delivery outcome [15, 16]. Furthermore, understanding pregnant women's intentions and related factors in rural settings is crucial [17].

Although numerous studies have been conducted on practices of facility delivery and their determinants, there is a gap in evidence regarding pregnant women's intentions, particularly in rural settings. Therefore, this study sought to address this gap by assessing pregnant women's intention and related factors to give birth in a healthcare setting in the Aleta-Wondo rural District, Ethiopia.

## Materials and methods

### Study setting and period

This study was conducted in the Aleta-Wondo rural District, one of 36 Districts in Sidama Regional State. It is 64 km far from Hawassa City, the capital of Sidama Regional State. The District has 37 kebeles (The smallest administrative unit of the Federal Democratic Republic of Ethiopia), and six Health Centers. It has a total population of 191,472, from which approximately 102,054 are females, with 44,039 of reproductive age. This study was conducted from January 10 to February 3, 2022.

### Study design, sample size determination and sampling procedure

A community-based cross-sectional study design was used. Pregnant women in the Aleta-Wondo rural district of Sidama Regional State were the source population, while pregnant women in the selected kebeles were the population under study. Among this population, pregnant women with a gestational age of four months and above who lived at least six months in the study area were included in this study analysis. However, those who were sick and unable to give a response during the data collection period were excluded.

The sample size was determined by using a single population proportion formula, at a 95% level of confidence and, a 5% margin of error, considering the intention of giving birth at a healthcare setting 52.1% [8] and 10% non-response rate. Accordingly, the final sample size was

421 pregnant women. The kebeles were selected by a lottery method after collecting all necessary information from the District administrative office. A preliminary survey was carried out in each of the selected kebeles one week before starting data collection to get the eligible women. All relevant information about the pregnant women was collected and cross-checked with the records of health posts.

Finally, the sample was proportionally allocated to each kebele based on the number of eligible women. The simple random sampling technique was employed to select study participants. For households with more than one eligible woman, one woman was selected by a lottery method.

### Data collection, analyzing, and data quality assurance procedures

Data were collected by using an interviewer-administered structured questionnaire. The first section of the tool contained the socio-demographic factors, obstetrics, and healthcare service-related characteristics of the study participants. The second section covered items designed to assess the perceived threats, perceived benefits, perceived barriers, and cues to action.

To assure data quality, the tool was prepared in English, translated into the local language, and back to English by translation specialists to verify the consistency of translation. Eight data collectors who had previous experience in data collection and were familiar with the local language (Sidaamu Afoo) were recruited to collect data. Four supervisors were assigned to control the overall activities of data collection. All data collectors and supervisors were trained for two days by the principal investigator before starting data collection. The training was given on the general objective of the study, the contents of the tool, and how to approach the study participants. A pre-test was conducted on the 5% of the total sample size outside of the study area to verify the efficiency of the tool, and any necessary amendments were done. In addition, data were manually cleaned and cross-checked for completeness before data entry.

The outcome variable for this study was pregnant women's intentions to give birth at a health facility. It was measured by asking the pregnant women about their plan regarding a preferred place of delivery for their current pregnancy (either home or health facility). The independent variables were socio-demographic, obstetric/reproductive, and health service-related characteristics of the study participants. To measure the participants' perceptions towards the benefits of giving birth at a health facility as well as complications related to pregnancy and childbirth, a Likert scale ranging from strongly disagree (1) to strongly agree (5) was used. Accordingly, 18 items were applied (5 items for perceived threats, 6 items for perceived benefits, 4 items for perceived barriers, and 3 items for cues to action). Finally, the items were summed up to produce a composite score, and the mean score was used for further analysis.

The collected data were entered into Epi Data version 3.1 and exported to SPSS version 21 for further analysis. Descriptive analysis and cross-tabulations were performed to see the distribution of predictor variables with the outcome variable. The goodness-of-fit of the model was also checked by Hosmer-Lemeshow goodness of model fit. Multicollinearity was checked among predictors. Bivariable analysis was done for each independent variable with the outcome variable, and variables with a p-value $<0.25$ were considered as candidates for multivariable logistic regression analysis to control possible confounders. Adjusted odds ratio (AOR) with 95% confidence interval (CI) was calculated to determine the presence and strength of association among predictors and the outcome variables. A P-value $\leq 0.05$ was used to consider statistical significance. Finally, the results were described by texts, figures, and tables.

### Ethical approval

This study was conducted with the approval of the Ethical Review Committee of the Yirgalem Hospital Medical College (Approval number: YHMC/IRB001). A written informed consent was obtained from the study participants after informing the aim of the study. An informed consent was also obtained from the parent/guardian of each participant under 18 years of age.

## Results

### Socio-demographic characteristics of the study participants

A total of 416 participants were included in this study, making a response rate of 98.8%. The mean age of the respondents was 24 (SD±4.34) years. About 159 (38.2%) of the respondents were within the age range of 20–24 years. Regarding religion and ethnicity, 345 (82.9%) and 366 (88%) were Protestants and Sidama respectively. Concerning educational status, 179 (43%) of the respondents attended primary education (Table 1).

### Pregnant women's intention to give birth at health facility, and other obstetric and health service related characteristics

Out of the respondents, about 255 [61.3% (95% CI: 57.0, 66.3)] intended to give birth at a health facility for their current pregnancy. Regarding the number of pregnancies, 174 women had 2–3 pregnancies. Two-thirds of the respondents reported that their index pregnancy was planned. About 271 (65.1%) of the respondents had information from health professionals. Concerning antenatal care service, 289 (69.5%) respondents started ANC follow-up for their last pregnancy (Table 2).

More than half (56.3%) of respondents noted that they perceived threats to delivery-related complications. The majority of pregnant women (89%) expressed that they perceived a benefit to giving birth at a health facility. Approximately two-thirds (n = 278) of the women included in the study reported they felt they experienced barriers to giving birth at a health facility.

The mean score was computed for each construct of the respondent's perceptions of the benefits of giving birth in a healthcare setting. Accordingly, the mean scores were 13.07 (SD ±2.870) for perceived threat, 12.73 (SD ±2.741) for perceived benefit, and 6.7 (SD 1±2.791) for perceived barriers (Fig 1).

### Factors associated with intentions of the pregnant women to give birth at a health facility

In the bivariable logistic regression analysis, the status of the current pregnancy, number of pregnancies, receiving information from health professionals, initiation of ANC follow-up, perceived threat, perceived benefits, perceived barriers, and cues to action were predictors of pregnant women's intention. In the multivariable logistic regression analysis receiving information from health professionals, perceived threats, perceived benefits, and perceived barriers were significantly associated with pregnant women's intentions to give birth at a health facility. Accordingly, the probability of preferring health facilities to give birth increased when pregnant women were exposed to health-related information. The likelihood of intent to deliver at a health facility was 2.6 times (AOR = 2.6; 95% CI: 1.5, 4.4) higher among pregnant women who received information from health professionals when compared with those who did not receive information.

Similarly, pregnant women who perceived threats towards delivery-related complications were 4.5 times (AOR = 4.5; 95% CI: 2.6, 7.6) more likely to intend to give birth at a health facility when compared with those who did not perceive threats. It was observed that there was a

**Table 1. Socio-demographic characteristics of the study participants in the Aleta-Wondo rural District, Ethiopia, 2022.**

| Variables | Category | Frequency | Percentage |
|---|---|---|---|
| Age of the woman | 15–19 years | 52 | 12.5 |
| | 20–24 years | 159 | 38.2 |
| | 25–29 years | 147 | 35.3 |
| | ≥30 years | 58 | 13.9 |
| Religion | Protestant | 345 | 82.9 |
| | Muslim | 44 | 10.6 |
| | Orthodox | 22 | 5.3 |
| | Others[1] | 5 | 1.2 |
| Ethnicity | Sidama | 366 | 88.0 |
| | Amhara | 21 | 5.0 |
| | Oromo | 19 | 4.6 |
| | Others[2] | 10 | 2.4 |
| Marital status | Single | 6 | 1.4 |
| | Married | 388 | 93.3 |
| | Divorced/widowed | 22 | 5.3 |
| Educational status of the women | No formal education | 131 | 31.5 |
| | Primary | 179 | 43.0 |
| | Secondary | 76 | 18.3 |
| | College and above | 30 | 7.2 |
| Occupation status of the women | Housewife | 278 | 66.8 |
| | Merchants | 89 | 21.4 |
| | Government employee | 32 | 7.7 |
| | Other[3] | 17 | 4.1 |
| Educational status of the husband | No formal education | 128 | 31.8 |
| | Primary | 136 | 33.8 |
| | Secondary | 89 | 22.1 |
| | College and above | 49 | 12.2 |
| Occupational status of the husband | Farmer | 194 | 48.3 |
| | Merchants | 128 | 31.8 |
| | Government employee | 62 | 15.4 |
| | Other[3] | 18 | 4.5 |

[1]Catholic

[2]Wolaita, Gurage

[3]Daily laborer, Carpenter

significant association between pregnant women's perceived benefits and their intention to give birth at a health facility. Accordingly, women who perceived the benefits of giving birth at a health facility were 2.3 (AOR = 2.3; 95% CI: 1.1, 4.9) times more likely to intend to deliver at a health facility. Additionally, we found a negative association between perceived barriers and the intention of the pregnant women included in this study to give birth at a health facility. Pregnant women who perceived barriers were 60% (AOR = 0.4; 95% CI: 0.3, 0.7) less likely to intend to give birth at a health facility (Table 3).

## Discussion

Births assisted by medical professionals in a healthcare setting are recognized as the best way to ensure safe and successful delivery outcomes for mother and baby [18]. Thus, this study

**Table 2. Obstetric and health service related characteristics of the study participants in the Aleta-Wondo rural District, Ethiopia, 2022.**

| Variables | Category | Frequency | Percentage |
|---|---|---|---|
| Number of pregnancy | 1 | 105 | 25.3 |
| | 2–3 | 174 | 41.8 |
| | ≥4 | 137 | 32.9 |
| Number of livebirths | 1 | 102 | 32.8 |
| | 2–3 | 145 | 46.6 |
| | ≥4 | 64 | 20.6 |
| Status of the last pregnancy | Unplanned | 101 | 24.3 |
| | Planned | 315 | 75.7 |
| Informed on place of delivery by health professionals | No | 145 | 34.9 |
| | Yes | 271 | 65.1 |
| Informed on delivery related complication | No | 145 | 34.9 |
| | Yes | 271 | 65.1 |
| Place of previous delivery | Home | 162 | 52.1 |
| | Health facility | 149 | 47.9 |
| Faced complications during previous delivery | No | 282 | 90.7 |
| | Yes | 29 | 9.3 |
| Types of complications faced | Delayed labor | 14 | 48.3 |
| | Fetal distress | 8 | 27.6 |
| | Vaginal bleeding | 5 | 17.2 |
| | Other | 2 | 6.9 |
| Initiated ANC follow-up for current pregnancy | No | 127 | 30.5 |
| | Yes | 289 | 69.5 |
| Time of initiating ANC follow-up | ≤16 weeks of gestation | 85 | 29.4 |
| | >16 weeks of gestation | 204 | 70.6 |
| Preferred place of delivery for current pregnancy | Home | 161 | 38.7 |
| | Health facility | 255 | 61.3 |

tried to assess the pregnant women's intention and related factors to give birth in a healthcare setting. Accordingly, the result of this study is comparable with the findings of previous study [19]. However, it was higher when compared with previous studies done in the North Gonder [8], Achefer district [20], Woldia district [21], Afar region [22], Eritrea [23], Uganda [24], Nigeria [25], Kenya [26] and Nepal [27]. Nevertheless, the result of this study was lower when compared with the previous studies done in Debremarkos Town [17], South-West Ethiopia [28], and Kenya [29]. The possible reason for this variation might be due to the differences in the study period and the improvement of the healthcare services. Furthermore, the difference in the study setting and target population could contribute to this variation. Unlike previously published studies, our study was conducted in rural areas with no nearby access to healthcare services. A knowledge gap among rural residents influences their intentions of giving birth in a health facility.

This study revealed that the pregnant women who received information from health professionals were more likely to intend to deliver at a health facility. This finding is in line with the previous studies done in Pakistan [16], Achefer district [20], and Nepal [27]. The possible explanation could be that creating awareness of the benefits of using the existing healthcare services could encourage pregnant women to prefer a healthcare setting to deliver their child.

This study showed that perceived threats towards childbirth-related complications increase the probability of giving birth in a health facility. This result is in agreement with previous

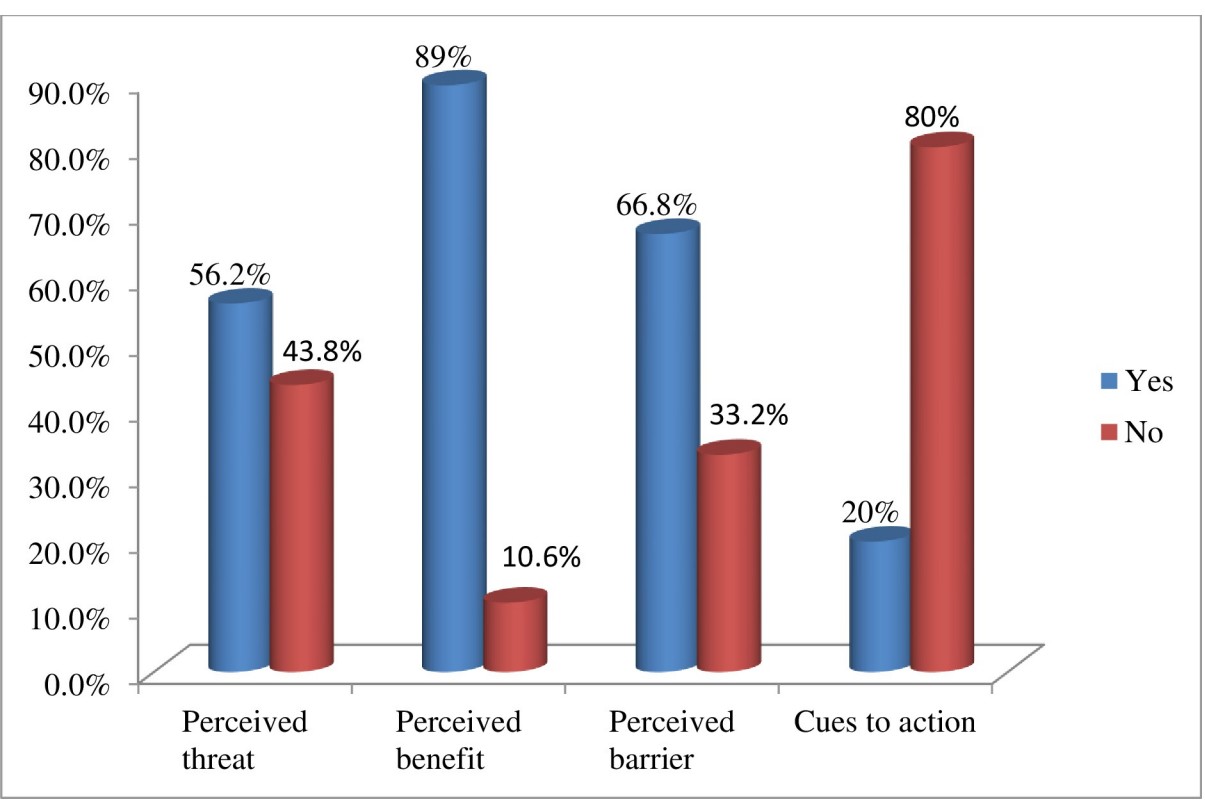

**Fig 1. The Perceptions of the study participants in the Aleta-Wondo Rural District, Ethiopia, 2022.**

findings from Debremarkos town [17], Achefer district [20], Afar region [22], and Jimma Zone [30]. When pregnant women are worried about complications related to childbirth, the likelihood to intend to delivery in a healthcare facility will be increased [27]. This study also found that the perceived benefits of giving birth in a healthcare setting increase the chance of giving birth in a health facility. This was in line with the previous studies done in the Debremarkos town [17], Afar region [22], and Jimma Zone [30]. Having a positive attitude towards the importance of the service could increase the interest to use it. Therefore, we can interpret that when the benefits overweight the risks, the chance of utilizing the service would probably be high.

Additionally, this study revealed a negative association between perceived barriers and pregnant women's intention to give birth in a healthcare setting. This finding is supported by previously published studies from the Achefer district [20], Afar region [22], Nepal [27], and Jimma Zone [30]. In rural settings, pregnant women face various obstacles to using healthcare services. As a result, they will be less likely to intend to use healthcare facilities to deliver their child.

## Limitations of the study

This study has some limitations; first, it does not confirm the absolute cause-and-effect relationship between the outcome variable and its predictors due to the cross-sectional nature of the study design. Second, some information was collected based on previous experiences of the respondents. Thus, the study may dispose to recall bias. Regardless of its limitations, this study has contributed new knowledge to the scientific community by assessing various factors

**Table 3. Bivariable and multivariable logistic regression analysis of factors associated with pregnant women's intentions to give birth at a health facility in the Aleta-Wondo rural district, Ethiopia, 2022.**

| Variables | Category | Preferred place of delivery | | COR with 95% CI | AOR with 95% CI |
|---|---|---|---|---|---|
| | | Home | Health facility | | |
| Status of current pregnancy | Unplanned | 58 | 43 | 1 | 1 |
| | Planned | 103 | 212 | 2.8 (1.8, 4.4) | 0.9 (0.5, 1.7) |
| Received information from health professionals | No | 97 | 48 | 1 | 1 |
| | Yes | 64 | 207 | 6.5 (4.2, 10.2) | 2.6 (1.5, 4.4)** |
| Started ANC | No | 70 | 57 | 1 | 1 |
| | Yes | 91 | 198 | 2.6 (1.7, 4.1) | 1.3 (0.8, 2.3) |
| Number of pregnancy | 1 | 41 | 64 | 1.4 (0.8, 2.3) | 0.9 (0.5, 1.7) |
| | 2–3 | 56 | 118 | 1.8 (1.2, 2.9) | 1.3 (0.8, 2.3) |
| | ≥4 | 64 | 73 | 1 | 1 |
| Perceived threat | No | 117 | 65 | 1 | 1 |
| | Yes | 44 | 190 | 7.8 (4.9, 12.1) | 4.5 (2.6, 7.6)** |
| Perceived benefit | No | 29 | 15 | 1 | 1 |
| | Yes | 132 | 240 | 3.5 (1.8, 6.8) | 2.3 (1.1, 4.9)** |
| Perceived barrier | No | 35 | 103 | 1 | 1 |
| | Yes | 126 | 152 | 0.4 (0.3, 0.6) | 0.4 (0.3, 0.7)* |
| Cues to action | No | 138 | 195 | 1 | 1 |
| | Yes | 23 | 60 | 1.8 (1.1, 3.1) | 1.1 (0.6, 2.1) |

**statistically significant at a p-value <0.001

*significant at a p-value <0.05

that affect pregnant women's intention to use healthcare settings for delivery, particularly in the rural setting of Ethiopia.

## Conclusions

This study revealed that less than two-thirds of pregnant women intended to give birth in a healthcare setting. Receiving information from health professionals, perceived benefits, perceived threats, and perceived barriers are factors significantly associated with pregnant women's intentions to give birth at a healthcare facility. Strengthening information communication with healthcare professionals and reducing threats and barriers that affect pregnant women's intentions is essential. Moreover, we recommend further research with mixed methods.

## Supporting information

**S1 Dataset. Dataset of the study.**
(SAV)

## Acknowledgments

We would like to thank Yirgalem Hospital Medical College for its support to conduct this study. We would like to extend our genuine gratitude to the data collectors, supervisors and study participants.

## Author Contributions

**Conceptualization:** Aregahegn Dona.

**Data curation:** Aregahegn Dona.

**Formal analysis:** Aregahegn Dona.

**Investigation:** Aregahegn Dona.

**Methodology:** Aregahegn Dona.

**Software:** Aregahegn Dona.

**Validation:** Aregahegn Dona.

**Visualization:** Azmach Dache Mue.

**Writing – original draft:** Aregahegn Dona.

**Writing – review & editing:** Aregahegn Dona, Azmach Dache Mue.

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
