## [Decision Letter · Decision Letter 0]

30 Jan 2024

PGPH-D-23-02489

The intentions of pregnant women to give birth at a health facility and associated factors in the Aleta-Wondo rural District, Ethiopia: A community based cross-sectional study

Dear Dr. Dona,

Thank you for submitting your manuscript to PLOS Global Public Health. After careful consideration, we feel that it has merit but does not fully meet PLOS Global Public Health’s publication criteria as it currently stands. Therefore, we invite you to submit a revised version of the manuscript that addresses the points raised during the review process.

We look forward to receiving your revised manuscript.

Kind regards,

Nicola Hawley

Academic Editor

Journal Requirements:

-doi:10.1136/bmjopen-2018-023013

-https://www.longdom.org/open-access-pdfs/perceptions-of-home-delivery-risk-and-associated-factors-among-pregnant-mothers-in-north-achefer-district-amhara-region-.pdf

- DOI: 10.1177/23333928211062777

3. In your revision ensure you cite all your sources (including your own works), and quote or rephrase any duplicated text outside the methods section. Further consideration is dependent on these concerns being addressed.

Additional Editor Comments (if provided):

Reviewers' comments:

Reviewer's Responses to Questions

**Comments to the Author**

1. Does this manuscript meet PLOS Global Public Health’s publication criteria? Is the manuscript technically sound, and do the data support the conclusions? The manuscript must describe methodologically and ethically rigorous research with conclusions that are appropriately drawn based on the data presented.

Reviewer #1: Partly

Reviewer #2: Yes

2. Has the statistical analysis been performed appropriately and rigorously?

Reviewer #1: Yes

Reviewer #2: I don't know

3. Have the authors made all data underlying the findings in their manuscript fully available (please refer to the Data Availability Statement at the start of the manuscript PDF file)?

Reviewer #1: No

Reviewer #2: Yes

4. Is the manuscript presented in an intelligible fashion and written in standard English?

Reviewer #1: No

Reviewer #2: Yes

5. Review Comments to the Author

Reviewer #1: Overall comments to the author:

- Significant grammar adjustments necessary for publication, currently tone does match the caliber of journal selected

- Flow seems very “piece-y”

- Important to provide additional clarity around methodology and why decisions were made

- Need for more specificity and clarity around presented results

- Do not present results in the discussion, need to use this section to tie results into context and larger body of existing work

- Need to strengthen conclusion

Reviewer #2: Reviewer comment to author

1. The study presents the results of primary scientific research. -yes

2. Results reported have not been published elsewhere.-yes

3. Experiments, statistics, and other analyses are performed to a high technical standard and are described in sufficient detail.-yes

4. Conclusions are presented in an appropriate fashion and are supported by the data-yes.

5. The article is presented in an intelligible fashion and is written in standard English-moderate/requires further modification.

6. The research meets all applicable standards for the ethics of experimentation and research integrity-yes/requires some details in ethical clearance regarding approval letter and types of the consent obtained.

7. The article adheres to appropriate reporting guidelines and community standards for data availability-yes/but authors must submit STROBE checklist for cross sectional study.

General comments

Unnecessary repetition of sentence regarding intention to give birth at health facility in abstract part line 22-23…..why?

Threat perception is emphasized in this model as a key step in distinguishing the importance of taking a recommended action. See line 61-62 page 4. Which model?

What was the research gap regarding intention of women to give birth at health facility? If any put your strong justification in final paragraph of the background.

Describe the characteristics of study population regarding institutional delivery and health facility availability/accessibility and other important information in your study setting. See page 5 line 82-85.

Specify study period (starting and end date). You said “This study was conducted from January to February 2022”. See page 5 line 85.

Revise this sentence: Whereas, the study population was pregnant women who were living in the randomly selected kebeles, and fulfilled

the eligibility criteria were. See Line 89-90

You said “An interviewer-administered, structured, and pretested questionnaire was developed by reviewing related literature to collect data”. What are those related literature? See line 107.

The dependent variables were Socio-demographic and Reproductive and health service related characteristics of the study participants. Revise it. See line 137-138

There was a problem of organizing your manuscript with appropriate sequence. For example you organized study variables and data quality control after data analysis. Revise it.

Have you checked multi-collinearity of independent variables?

Approval number at which responsible individual in college provided?

Page 9... mention others for religion and ethnicity

If you were intended to use health belief model why you missed other constructs of this model?

You said “Facing different challenges would affect pregnant woman’s decision-making ability to use the available services. Furthermore, there are numerous socio-cultural and economic problems”. See discussion part line 261. But you haven’t addressed economic related factors (income/wealth index of household) and cultural factors (qualitative study). ..

State your strength of study if any

Some references are too old (see reference number 2,3 10,11,19)

Overall comment

Manuscript was well written and the problem under the study was interesting public health issue. My concern was justification for this study because there were studies done regarding this issue. Some editorial, topographic and description problem should be corrected.

Final decision- Accept with minor modification.

6. PLOS authors have the option to publish the peer review history of their article (what does this mean?). If published, this will include your full peer review and any attached files.

**Do you want your identity to be public for this peer review?** For information about this choice, including consent withdrawal, please see our Privacy Policy.

Reviewer #1: No

Reviewer #2: **Yes: **Elias Amaje Hadona

---

## [Decision Letter · Decision Letter 1]

18 Apr 2024

The intentions of pregnant women to give birth at a health facility and associated factors in the Aleta-Wondo rural District, Ethiopia: A community based cross-sectional study

PGPH-D-23-02489R1

Dear Mr. Dona,

We are pleased to inform you that your manuscript 'The intentions of pregnant women to give birth at a health facility and associated factors in the Aleta-Wondo rural District, Ethiopia: A community based cross-sectional study' has been provisionally accepted for publication in PLOS Global Public Health.

Best regards,

Nicola Hawley

Academic Editor

Reviewer Comments (if any, and for reference):

Reviewer's Responses to Questions

**Comments to the Author**

1. If the authors have adequately addressed your comments raised in a previous round of review and you feel that this manuscript is now acceptable for publication, you may indicate that here to bypass the “Comments to the Author” section, enter your conflict of interest statement in the “Confidential to Editor” section, and submit your "Accept" recommendation.

Reviewer #1: All comments have been addressed

Reviewer #2: All comments have been addressed

2. Does this manuscript meet PLOS Global Public Health’s publication criteria? Is the manuscript technically sound, and do the data support the conclusions? The manuscript must describe methodologically and ethically rigorous research with conclusions that are appropriately drawn based on the data presented.

Reviewer #1: Yes

Reviewer #2: Yes

3. Has the statistical analysis been performed appropriately and rigorously?

Reviewer #1: Yes

Reviewer #2: Yes

4. Have the authors made all data underlying the findings in their manuscript fully available (please refer to the Data Availability Statement at the start of the manuscript PDF file)?

Reviewer #1: Yes

Reviewer #2: Yes

5. Is the manuscript presented in an intelligible fashion and written in standard English?

Reviewer #1: Yes

Reviewer #2: Yes

6. Review Comments to the Author

Reviewer #1: ABSTRACT

• L28/29 – need a stronger last sentence of your abstract, currently quite weak

INTRO

• L34/35 – combine into one paragraph

• L37 – “women’s intention to choose” is quite awkward, suggest rephrase to “women’s choices for birth”

• L39 – this sentence seems a bit jarring as there’s no lead in; maybe swap the second and first sentences

• L43 – it’s unclear what you mean by “intentions” consider rephrasing, wider sentence is a bit unclear

• L46 – delete “for low-income countries”

• L48 – “ignorance of receiving skilled care” is both a bit unclear and feels quite accusatory, consider rephrase

• L51 – intentions for what? It’s important that you clarify exactly what you mean

• L54 – is that a proven fact? If not, proposed rephrase: “pregnant women…are often more interested…”

• L58/59 – the connection between these sentences is not clear. Suggest adding an additional sentence to better make the connection

MATERIALS & METHODS

• L74 – why was this methodology used? Suggest adding a sentence justifying your choice

• L84 – what preliminary information was collected? Why was this collected? Suggest adding this in as a supplementary document

• L125 – delete “finally, the results were described by texts, figures and tables”

RESULTS

• L129 – suggest changing “making a response rate” to “with a response rate”

• L130 – “about” 159 respondents doesn’t make sense; was it 159 or was that an approximation?

• L139 – again, don’t use the word “about” if it’s not an approximation

• L140/141 – suggested rephrase: Approximately X% (n=174) women reported having previously experienced 2-3 pregnancies.

• L142 – again, don’t use the word “about” if it’s not an approximation

• L142/143 – suggested rephrase: Approximately 70% (n=289) women accessed antenatal care services during the index pregnancy.

• L146 – suggest adding: The VAST majority

• L149-152 – would suggest providing min/max scores possible as well as further interpreting the score results as at present they don’t add much to this section

• L162 – add a comma after “analysis”

DISCUSSION

• L188/189 – consider combining these sentence or providing more clarification

• Suggest changing the order of how the discussion is presented to present your study’s findings first and the comparison to other studies after

• Is the point of the inclusion of all of the comparisons to the other settings to show that the findings can be extrapolated to other contexts / are extra relevant? If so, please clarify. At the moment it seems to detract from your overall message.

• L219 – change “some” to “several”

• L221 – what does “some” mean here? Clarify

• L221/222 – combine these two sentences and consider rephrase: “therefore the study may reflect some recall bias”

Reviewer #2: all comments were addressed adequately.

7. PLOS authors have the option to publish the peer review history of their article (what does this mean?). If published, this will include your full peer review and any attached files.

**Do you want your identity to be public for this peer review?** For information about this choice, including consent withdrawal, please see our Privacy Policy.

Reviewer #1: **Yes: **Hayley Conyers

Reviewer #2: **Yes: **Elias Amaje Hadona
